# Formulation of Lipid-Based Nanoparticles for Simultaneous Delivery of Lapatinib and Anti-Survivin siRNA for HER2+ Breast Cancer Treatment

**DOI:** 10.3390/ph15121452

**Published:** 2022-11-23

**Authors:** Sahar Eljack, Stephanie David, Igor Chourpa, Areeg Faggad, Emilie Allard-Vannier

**Affiliations:** 1EA6295 Nanomédicaments et Nanosondes (NMNS), University of Tours, 37020 Tours, France; 2Department of Pharmaceutics, Faculty of Pharmacy, University of Gezira, Wad Medani 21111, Sudan; 3Department of Molecular Biology, National Cancer Institute, University of Gezira (NCI-UG), Wad Medani 21111, Sudan

**Keywords:** lipid-based nanoparticles, co-delivery, Lapatinib, Survivin, siRNA, HER2+ breast cancer

## Abstract

In this work, lipid-based nanoparticles (LBNP) were designed to combine tyrosine kinase inhibitor (TKI) Lapatinib (LAPA) with siRNA directed against apoptosis inhibitor protein Survivin (siSurvivin) in an injectable form. This nanosystem is based on lipid nanocapsules (LNCs) coated with a cationic polymeric shell composed of chitosan grafted through a transacylation reaction. The hydrophobic LAPA is solubilized in the inner oily core, while hydrophilic siRNA is associated electrostatically onto the nanocarrier’s surface. The co-loaded LBNP showed a narrow size distribution (polydispersity index (PDI) < 0.3), a size of 130 nm, and a slightly positive zeta potential (+21 mV). LAPA and siRNA were loaded in LBNP at a high rate of >90% (10.6 mM) and 100% (4.6 µM), respectively. The siRNA-LAPA_LBNP was readily uptaken by the human epidermal growth factor receptor 2 overexpressed (HER2+) breast cancer cell line SK-BR-3. Moreover, the cytotoxicity studies confirmed that the blank chitosan decorated LBNP is not toxic to the cells with the tested concentrations, which correspond to LAPA concentrations from 1 to 10 µM, at different incubation times up to 96 h. Furthermore, siCtrl.-LAPA_LBNP had a more cytotoxic effect than Lapatinib salt, while siSurvivin-LAPA_LBNP had a significant synergistic cytotoxic effect compared to siCtrl.-LAPA_LBNP. All these findings suggested that the developed modified LBNP could potentiate anti-Survivin siRNA and LAPA anti-cancer activity.

## 1. Introduction

Globally, cancer has become the second leading cause of death. Women’s breast cancer has now exceeded lung cancer as the leading cause of worldwide cancer. In 2020, there were an estimated 2.3 million new cases, representing 11.7% of all cancer cases. It is the fifth leading cause of cancer mortality worldwide, with 685,000 deaths per year. Among women, breast cancer accounts for 1 in 4 cancer cases and 1 in 6 cancer deaths [1]. The most common receptors that are overexpressed in breast cancer cells are part of the epidermal growth factor receptor (EGFR) family of receptor tyrosine kinases. EGFR and human epidermal growth factor receptor-2 (HER2) are overexpressed in approximately 40% and 25% of breast cancers, respectively, and are associated with an aggressive tumor nature and poor prognosis [2]. Fortunately, in recent years, HER2+ breast cancer has been regarded as a curable illness with a highly hopeful therapeutic outcome due to the emergence of targeted therapies directed against HER2 receptors, such as monoclonal antibodies (Mabs) (e.g., trastuzumab, pertuzumab), antibody-drug conjugates (ADCs) (e.g., trastuzumab emtansine, trastuzumab deruxtecan), and tyrosine kinase inhibitors (TKIs) (e.g., lapatinib, neratinib, tucatinib, and erlotinib) [3,4,5,6,7]. Lapatinib, a dual EGFR/HER2 kinase inhibitor, has been approved for oral use in patients with trastuzumab-refractory HER2-overexpressing breast cancer since 2007 [8]. Unfortunately, Lapatinib is categorized in the biopharmaceutics classification system (BCS) as a class-II drug (low solubility, high permeability) and possesses suboptimal pharmacokinetic properties. Its oral bioavailability has considerable interpatient variability and is significantly affected by food intake [8,9]. It is also evident from many research works that both low-fat and high-fat diets before LAPA administration increase their bioavailability [8,9]. Besides food intake, gastric pH also influences LAPA bioavailability. Higher gastric pH reduces LAPA absorption; concomitant use of drugs interfering with gastric acidity reduces LAPA bioavailability [9,10]. Lapatinib is also extensively bound to plasma albumin and alpha-1 glycoprotein, reducing its systemic exposure at the tumor sites. Because of its low water solubility, poor biostability, and resistance risks, as well as concerns with patient compliance (dose of several pills per day), its clinical use is limited [11,12]. One alternative could be the formulation of a vectorized form of Lapatinib that can be injected via an intravenous route.

Current research has also demonstrated that combinations of anti-cancer drugs and gene modulatory therapies provide better therapeutic outcomes compared to monotherapy [13]. This type of combination could potentially increase (i) the synergistic action of each therapeutic modality and (ii) the tumor site selectivity, while (iii) effectively reversing the drug resistance [14]. In this case, Lapatinib will be combined with anti-survivin siRNA (siSurvivin). LAPA induces its effect through tyrosine kinase receptor inhibition and provides additive downregulation of anti-apoptotic genes such as Survivin [15]. Indeed, Survivin is one of the inhibitors of apoptosis proteins with a bi-functional role in regulating cell growth and inhibiting apoptosis. Thus, silencing Survivin expression could be a viable way to trigger cell death in tumor sites [16,17,18,19].

To date, many drug delivery systems have been described and optimized to maintain the co-delivery of different therapeutics, including polymer-based nanoparticles [20], lipid-based nanoparticles [21], and hybrid inorganic nanoparticles [22]. These approaches hold great promise in cancer therapeutics by reducing side effects and improving therapeutic efficacy, tumor targeting, and reversal of drug resistance status along with synergistic potentiated therapeutic outcomes. Because of their unique characteristics, lipid nanocapsules (LNCs) have gained much attention among the various drug-delivery systems. Their formulation is based on the phase-inversion method. Their structure consists of an oily core surrounded by a shell composed of a mixture of lecithin and hydroxy stearate of polyethylene glycol [23]. LNC formulations have exciting features: their main ingredients have been approved for in vivo intravenous administration, and their development method is simple and uncomplicated without further addition of toxic or organic solvents required for high drug loading capacities. Moreover, LNCs are versatile nanoparticles that can undergo further modifications to enhance their stability and specificity, such as (i) surface functionalization, (ii) the ability to incorporate different solubilizing agents to enhance the encapsulation efficiency of drugs with low solubility profiles, and (iii) their ability to be combined with other types of treatments [24,25,26,27,28]. Furthermore, it has been demonstrated that LNCs can be used for the single delivery of both siRNA, and lipophilic chemotherapeutics [29,30,31].

Our present study provides a detailed, comprehensive approach for developing a parenteral form of Lapatinib and anti-Survivin siRNA as a treatment choice for HER2+ breast cancer. Multifunctional lipid-based nanoparticles (LBNP) were designed to co-deliver LAPA and anti-Survivin siRNA to potentiate LAPA’s therapeutic effect. For this purpose, LAPA-loaded LNCs (LAPA_LNCs) were prepared to encapsulate LAPA in the oily core using the solubility enhancer Labrasol^®^, followed by surface modification of LAPA_LNCs with chitosan through a transacylation reaction between the pegylated hydroxy stearate and the functional amino groups of chitosan to develop LAPA_LBNP [32,33]. Afterward, negatively charged siRNA was loaded via electrostatic interaction on the positively charged LBNP to obtain siRNA-LAPA_LBNP. The obtained LBNPs have been characterized by determining their size, polydispersity index (PDI), surface charge, LAPA encapsulation efficiency, siRNA complexation, and storage stability. Then, in vitro biological evaluation was carried out to study their cellular uptake and cytotoxicity.

## 2. Results and Discussion

### 2.1. Formulation of LAPA_LBNP

LAPA_LNCs were prepared by a phase inversion temperature method [23,25]. The formulation process is schematically represented in Figure 1. Briefly, Lapatinib was dissolved in Labrasol^®^ as a solubility enhancer, then Labrafac^®^ WL 1349, Kolliphor^®^ HS15, Lipoid^®^ S75-3, NaCl, and water were added. Three progressive heating and cooling cycles were carried out between 85 °C and 45 °C. At the inversion phase (61–63 °C) of the last cooling cycle, 2 °C deionized water was added to the mixture. Afterwards, LAPA_LNCs underwent surface modification by a transacylation reaction to graft low molecular weight chitosan oligosaccharide (5 kDa) to obtain LAPA_LBNP [33]. Coating the nanoparticles with the cationic chitosan polymer allowed siRNA adsorption via electrostatic interactions, resulting in the formation of siRNA-LAPA_LBNP (Figure 1). Lapatinib is a small molecule with a very low water solubility at 25 °C (7 µg/mL) that was first dissolved with the help of a solubility enhancer Labrasol^®^, in order to be integrated into the LNC’s core. Labrasol^®^ is a self-emulsifying lipid-based excipient capable of forming microemulsions by simple stirring when it is brought into contact with water. However, its efficiency in generating such microemulsions depends on the type and the concentration of co-surfactants and oils used [25,34,35]. According to Libster et al., the presence of Labrasol^®^ in the lipid-water system can cause reorganization in the structure of formed liquid crystals, which is explained by a decrease in elasticity of the system due to the location of Labrasol^®^ at the interface and its ability to bind water molecules [36]. The final formulations of LNCs and LAPA_LNCs were opalescent white and faint yellow, respectively, with bluish reflection due to the Tyndall effect. Chitosan polymer was extensively studied in nanotechnology due to its high transfection efficiency and low toxicity compared to other polymers, such as polyethyleneimine (PEI), making it a favorable option in many cases. Its major hurdle is related to its poor solubility in neutral conditions and the production of large-sized nanoparticles when a high molecular weight form of chitosan is used [37]. Recent studies have shown that nano-delivery systems utilizing chitosan can help to overcome barriers related to the delivery of the medicine to the target, hence increasing the drug’s therapeutic efficacy [38]. Chitosan oligosaccharide lactate is a byproduct of chitosan degradation that has lately gained popularity due to its enhanced water solubility and suitability as a gene carrier [39,40,41]. Due to the nature of the amine groups, chitosan shows positive charges depending on the solvent pH, thus allowing electrostatic interactions with negatively charged molecules such as nucleic acids to form stable complexes.

### 2.2. Optimization of Chitosan LBNP

Our aim was to design LBNP with reasonable physicochemical properties that can co-load a high payload of both siRNA and LAPA, which is favorable for further cellular uptake and better cytotoxicity. siRNA complexation into LBNP with three different chitosan concentrations, referred to as 1×, 2×, and 3×, was evaluated. All formulations had sizes between 120 and 150 nm and a positive surface charge between +20 and +30 mV (Appendix A). Moreover, the size and PDI measurement values were close to each other, whereas the zeta potential values fluctuated over 28 days of the experimental period. In the 3× chitosan formulation, zeta potential values showed a declining trend between day 0 and day 28, with a zeta value reduction from ~+30 mV to +10 mV (Appendix A). The result of the agarose gel electrophoresis is shown in Figure 2. The three siRNA_LBNP formulations (chitosan 1×, 2×, and 3×, siRNA concentration 4.6 µM) were compared to free siRNA, and each sample was analyzed in the absence and presence of heparin in order to estimate the free and total siRNA amount in the formulations, respectively. The 1× and 2× LBNP formulations could not efficiently complex the entire amount of siRNA as there is a visible fluorescent band in the absence of heparin. As evident, the 3× LBNP formulation could complex the siRNA sufficiently, as almost no fluorescence is visible in the absence of heparin. The result showed that 1.8 mg/mL of chitosan (corresponding to 3×) in the formulation would result in an almost complete siRNA complexation. In the presence of heparin, all formulations showed equally intense fluorescence bands compared to free siRNA. This indicates that there was no siRNA alteration during the formulation process. According to these results, we chose a 3× LBNP formulation for the subsequent physicochemical characterization and cellular evaluations. 

### 2.3. Characterization of LBNP 

According to the dynamic light scattering measurements (Table 1), the average particle size of blank LNCs was 86.9 ± 12.9 nm, with a narrow PDI of 0.116 ± 0.023. The zeta potential of the blank LNCs was around −4.15 ± 4.35 mV. The surface charge of blank LNCs bears a neutral or moderate negative surface charge due to the presence of the PEG shell offered by Kolliphor^®^HS15 [36]. After surface modification by the chitosan layer, purification based on dialysis, and pH adjustment to load siRNA, the size of the LAPA_LBNP increases to 126.9 ± 20.6 nm, with a PDI of 0.145 ± 0.080, and a zeta potential of +28.42 ± 6.69 mV. The size increment explains the effective chitosan grafting at the surface of the NPs. Chitosan is a natural linear polymer of glucosamine/acetylglucosamine that behaves as a polyelectrolyte with positive charge density at low pH and acts as a pH-responsive polymer due to the protonation-deprotonation balance of its amino groups in different pH values [42,43,44]. The positive surface charge is due to their protonation in acidic media. siRNA loading reduced the size of co-loaded nanoparticles to 123.94 ± 17.10 nm, indicating that the siRNA integration in the chitosan layer resulted in a slight non-significant shrinkage in the NP’s size, with a uniform mono disperse distribution around 0.09 ± 0.052. A lower PDI of less than 0.1 was successfully obtained, indicating a high monodispersity. This was interesting, as this low value is not usually achievable in formulations using electrostatic self-assembly. As predicted, the surface charge of siRNA-LAPA_LBNP nanoparticles decreased to +20.84 ± 8.67 mV, indicating successful electrostatic loading between the polymer positive amino groups and siRNA negative phosphate groups (Table 1). All the formulations had a PDI of less than 0.3, which confirmed reasonable colloidal stability.

LAPA encapsulation efficiency studies were made after different formulation steps to ensure that the LAPA concentration is maintained during the whole formulation process of LBNP. As presented in Table 1, LAPA_LBNP had an EE of 94.51 ± 6.63%, with an experimental drug payload of 5.5–6.3 mg of LAPA per gram of LNC suspension (9.5 mM−10.9 mM). This very high loading capacity of a poorly soluble drug such as LAPA shows that the formulation has potential for parenteral administration. Buss et al. developed micelles with a total lapatinib content of 98.77 ± 2.01% relative to the theoretical value of 0.247 ± 0.005 mg/mL [45]. 

Afterwards, siRNA_LBNP and siRNA-LAPA_LBNP (prepared with 3× chitosan and siRNA concentration 4.6 µM) were assessed by gel electrophoresis (Figure 3). Free siRNA shows an intense fluorescence signal in the presence and absence of heparin and acts as a control. LBNP without siRNA showed no fluorescence signal as there was no free siRNA. However, there were two slight or intense fluorescence bands on top of the gel for blank LBNP and LAPA_LBNP, respectively, due to the presence of LBNP and LAPA. Many reports demonstrated that LAPA could function as a ‘turn-on’ fluorophore, as the hydrophobicity of LAPA leads to fluorescent aggregates in solution. Interaction between the lipid carrier and proteins produces a spectroscopically distinct photoemission that can be detected by UV [46]. siRNA_LBNP and siRNA-LAPA_LBNP show intense fluorescence bands at the siRNA level in the presence of heparin and a slight fluorescence signal in the absence of heparin, indicating some free siRNA in the formulation. The fluorescence signal at LAPA and LBNP levels was in concordance with the results obtained for blank LBNP and LAPA_LBNP. These results show that siRNA_LBNP and siRNA-LAPA_LBNP could ultimately entrap the siRNA. Moreover, LAPA encapsulation did not disturb the siRNA complexation into LBNP. Our LBNPs were able to load up to 4.6 µM of the siRNA into their polymer layer.

In brief, nanoparticles with a size of about 125 nm and a positive surface charge were produced. It is well-studied that both size and surface charge play a vital role in the nanoparticle fate, such as potential in vivo interaction with the negatively charged tumor cell membrane, serum stability, and, most importantly, charge-mediated cellular uptake [47,48,49,50]. For such effective penetration, the particle diameter should ideally reside between 10–150 nm as it will sustain a longer circulation time and increased accumulation in the target site. Moreover, positively charged NPs show better uptake by direct permeation than neutral and negatively charged NPs [51]. The NPs in the current study have physicochemical characteristics suitable for intravenous administration and could encapsulate LAPA in the core interface for efficiency up to more than 90% (~6.3 mg/g; 10.6 mM) and siRNA on the surface for up to 100% (4.6 µM). Moreover, the positive surface charge can enhance NP cellular delivery through charge-mediated uptake.

### 2.4. Storage Stability of siRNA_LBNP

Storage stability was investigated throughout the study period of 28 days. Physical appearance, size, PDI, and zeta potential values should be stable during the study period, indicating that the original nanoparticles did not undergo significant variations, kept their dimensions and homogeneity, and kept their surface modification properties. The physical and storage stability of the nanoparticulate system is essential for its potential in vivo application and the prediction of the nanoparticles’ fate inside the biological system [52]. Therefore, two distinct experiments were performed. The first one was in order to check LBNP stability and whether the siRNA loading is dependent on the age of the LBNP formulation; the second one compared the characteristics of siRNA_LBNP with a physical mixture of chitosan and siRNA (=polyplexes) over time. Optically, the formulations appeared the same, with no apparent signs of instability, such as flocculation or coalescence, for both experiments. 

For the first experiment, LBNP was prepared using an optimized chitosan concentration of 1.8 mg/mL, and siRNA was added at different time points just before the characterization (Figure 4 and Appendix A). Particle sizes remained constant over time, with low PDI values (Figure 4A). Zeta potential values (Figure 4B) were stable for 7 days and then dropped from day 7 to day 28 without significant siRNA release (Appendix A). The results indicate that the loading of siRNA in the first 7 days after LBNP formulation can be done at any moment. The surface charge decline between day 7 and day 28 could be explained by the pH change of the suspension over time, influencing the ionic strength or any chemical balance in the suspension [53,54]. The stability of LNCs is mainly due to steric hindrance provided by the pegylated surface. Blank unmodified LNCs showed stability in terms of size, PDI, and zeta potential values for more than six months of storage with very low fluctuation of the measured values (data not shown). LBNP formulations showed higher fluctuations of the measured values than LNCs. This could be due to the displacement of the PEG moieties by chitosan polymer after the transacylation reaction. 

For the second experiment, siCtrl._LBNP, and siCtrl._chitosan polyplex were prepared and characterized at different time points (Figure 5 and Figure 6). Like the polyplexes, the siCtrl._LBNP could entrap siRNA for the whole study period without any significant change (Figure 5) as there was almost no free siRNA visible in the conditions with heparin. Size, PDI, and zeta potential measurements were stable over 28 days for siCtrl._LBNP with low fluctuations in the values. This is probably due to the siRNA loading on LBNP, which renders the system more stable. siCtrl._LBNP had sizes less than 150 nm, PDI less than 0.2, and zeta potential around +30 mV. In contrast, the siCtrl._chitosan polyplexes had a decrease in size over time with variable PDI and zeta potential values (Figure 6). These results underline that siCtrl._LBNPs are more stable over time than siCtrl._chitosan polyplexes and that the grafting of chitosan at the surface of the LBNP stabilizes the complex. The formulations of siRNA_LBNP could be described as stable for up to 28 days. It is noteworthy to state that we did not go beyond the period of 28 days.

### 2.5. Cellular Uptake of Fluorescent siRNA_LBNP

The cellular uptake of blank LBNP and siRNA_LBNP on SK-BR-3, a HER2+ breast cancer cell line, was evaluated using confocal spectral imaging (CSI). A fluorescent dye (1,1-Dioctadecyl-3,3,3,3-tetramethylindodicarbocyanine, 4-chlorobenzenesulfonate salt DiD) was encapsulated in the core of LBNP instead of LAPA, and fluorescent ATTO488 labeled siRNA was used to follow LBNP and siRNA intracellular internalization, respectively. DiD_LBNP and ATTO488 labeled siCtrl.-DiD_LBNP were characterized to conduct the cellular uptake investigation (Appendix A). DiD is a lipophilic carbocyanine dye that appeared in the formulation as bright pink color, indicated by the red spectra in Figure 7A, with a maximum emission wavelength of 667 nm. ATTO 448 labeled siRNA-DiD_LBNP were also characterized for their size and PDI (Appendix A). ATTO 448 (green spectra, Figure 7A) has a maximum emission wavelength of 524 nm. The fluorescence signal was visualized inside the SK-BR-3 cells after 4 h incubation with the dyes-loaded LBNPs. In Figure 7B, the ATTO 488 labeled siRNA (green) was localized in the cytoplasm and perinuclear space. At the same time, the DiD_LBNP (red) was also mainly localized in the cytoplasm but not precisely at the exact location (Figure 7C). We hypothesized that a successful endosomal escape of siRNA was assured as the two fluorescences were not totally co-localized (Figure 7D). Our findings confirmed that chitosan modification did not interfere with LBNPs’ rapid cellular uptake. The loaded siRNA could penetrate into the cells and escape the endosomes allowing its RNA interference activity, as shown by the Survivin protein downregulation using Western blot (Appendix A). Significantly, enhanced cellular uptake of nanoparticles would undoubtedly facilitate the therapeutic effect of the loaded active contents, thereby improving the therapeutic efficacy of the drugs. The fluorescence signal appeared mainly inside the cells and not at the cell membrane, confirming the effective cellular uptake of LBNP. 

### 2.6. Cell Viability Analysis of LAPA_LBNP and siSurvivin-LAPA_LBNP

The cytotoxic effect of different LBNP formulations was assessed on SK-BR-3 breast cancer cells using the ATP-based cytotoxicity assay cell titer Glo^®^. As illustrated in Figure 8, the curves showed that all the formulations exhibited typical time and concentration-dependent cytotoxicity. The IC_50_ values extracted from the cell survival curves for blank LBNP, LAPA salt, siCtrl.-LAPA_LBNP, and siSurv.-LAPA_LBNP are presented in Table 2. Blank LBNP IC_50_ is considered non-toxic compared to all the formulations used, as the IC_50_ value was superior to 6 µM (*p* < 0.05) (Table 3). These results confirm that blank_LBNPs at the same range of concentration as loaded ones were not toxic to the cells, proving that the toxicity is more related to the encapsulated active contents than the carrier.

On the other hand, the toxicity of siCtrl.-LAPA_LBNP was statistically significant compared to free LAPA (*p* = 0.0006). The IC_50_ was around 99.7 ± 12.8 for siCtrl.-LAPA_LBNP and around 159.0 ± 12.4 for LAPA ditosylate. Higher cytotoxicity is related to more drugs transferred to cancer cells. Free drugs are available by a passive diffusion mechanism at the intracellular site. It is also worth mentioning that LAPA is a substrate of the ATP-dependent pump transporter system, mainly P-gp and ABCG2 [55,56]. It means that LAPA is pumped out of the cytoplasm before exerting its therapeutic efficacy [57]. On the contrary, drug-loaded nanoparticles had an apparent cytotoxicity due to the nanoscale effect with a mechanism of cellular internalization by endocytosis [58]. In our case, siCtrl.-LAPA_LBNP was more efficacious than free LAPA ditosylate, which is a very interesting result. Many other drug-loaded nanoparticles reported less potent activity because the encapsulated drug needed to diffuse through the core of the nanoparticles and then reach the cytoplasm compartment, while the free drug is easily accessible to their sites of action [59]. Although Lapatinib had a dual inhibition of HER2 and EGFR against overexpressed HER2 breast cancer, the drug cannot completely inhibit cell proliferation in the HER2+ cell line model. We conclude that the siCtrl.-LAPA_LBNPs had a better effect than the Lapatinib salt, which provides an excellent alternative parenteral dosage form of Lapatinib.

In addition, siSurvivin-LAPA_LBNP showed a more substantial effect on decreasing cell viability compared to siCtrl.-LAPA_LBNP, indicating that anti-Survivin siRNA and LAPA had synergistic anti-cancer effects (Figure 8 and Table 3). However, this synergy was marginal (*p* = 0.0418), meaning that further optimization must be carried out to achieve the maximal effect of the combination. The superior cytotoxic effect of siSurvivin-LAPA_LBNP was attributed to the additive effect of the downregulation of Survivin due to siRNA and LAPA co-vectorization. Xia et al. studied the relationship between Survivin protein downregulation and Lapatinib usage in HER2 overexpressing tumors. They concluded that selective knockdown of HER2 using small interfering RNA markedly reduced Survivin protein, resulting in apoptosis of HER2-overexpressing breast cancer cell lines such as BT-474. Alternatively, at relevant concentrations, inhibition of ErbB2 signaling using Lapatinib, a reversible HER2/EGFR tyrosine kinase inhibitor, leads to marked inhibition of Survivin protein resulting in cell apoptosis. The effect of Lapatinib on Survivin seems to be predominantly post-translational [60]. If the knockdown of the Survivin protein plays a role in lapatinib-induced apoptosis, then Survivin overexpression might protect cells from the therapeutic action of Lapatinib. Moreover, if the regulation of Survivin by Lapatinib is solely transcriptionally mediated, then Lapatinib would not be expected to reduce His-tagged Survivin protein, which is under the transcription of another promotor. Their finding provides a rationale for combining Lapatinib with small interfering RNA regulating apoptosis, leading to apparent cell death [60]. 

## 3. Materials and Methods

### 3.1. Materials 

Anhydrous dimethyl sulfoxide (DMSO), sodium chloride, and Tetrahydrofuran (THF) was purchased from Carlo Erba reagents (Normandie, France). Glycine was provided by Sigma Aldrich (Saint-Quentin-Fallavier, France). Kolliphor HS15^®^ (Macrogol 15-Hydroxystearate), and Labrafac^®^ WL 1349 (triglyceride with 50–80% caprylic acid and 20–50% capric acid, and the macrogol glycerides) were obtained from BASF (Ludwigshafen, Germany). Labrasol^®^ (30% of mono-, di- and triglycerides of C8 and C10 fatty acids, 50% of mono and di-esters of PEG, 20% of free PEG 400) were kindly provided by Gattefosse S.A. (Saint-Priest, France). Lapatinib (Mw = 581.1 g/mol) was provided by Acros Organic (France) and Lipoid^®^ S75-3 by Lipoïd (Lipoid GmbH, Ludwigshafen, Germany). DiD fluorescent dye (1,10-dioctadecyl-3,3,30,30-tetramethylindodicarbocyanine, 4-chlorobenzenesulfonate salt) was provided by Invitrogen^™^ (Waltham, MA, USA). Chitosan oligosaccharide lactate (MW 5000; degree of acetylation: ≤90%) was obtained from Sigma–Aldrich Chemie GmbH (Schnelldorf, Germany). Dialysis membranes (molecular weight cut-off (MWCO) 100 kDa, regenerated cellulose) were purchased from BioValley (Marne La Vallée, France). The cell titer Glo^®^ kit was obtained from Promega (Madison, WI, USA). Model siRNA (sense sequence 5′-’3′ GGAAGAUCAUAAUGGACAGdTdT with lower case letters representing deoxyribonucleotides) both labeled with ATTO488 (at ‘5’ position of the sense strand) or unlabeled were purchased from Eurogentec (Angers, France). siRNA against Survivin was purchased from Sigma Aldrich Chemie GmbH (St.Quentin Fallavier, France), sense sequence 5′-’3′ GUCUGGACCUCAUGUUGUUdTdT with lower case letters representing deoxyribonucleotides). For gel retardation assays, loading buffer, agarose, and ethidium bromide were from Fisher Bioreagents^®^ (Illkirch, France). Life Technologies (Paisley, UK) supplied all the culture media and supplements for cell culture. Water was obtained from a Milli-Q system (Millipore, Paris, France).

### 3.2. Nanocarrier Preparation

#### 3.2.1. Formulation of LAPA-Loaded Lipid Nanocapsules

LAPA-loaded lipid nanocapsules (LAPA_LNCs) were prepared according to the phase inversion temperature method described by Malzert-Fréon et al., with slight modifications [25]. The active was encapsulated in the core of the nanoparticles containing a mixture of a solubility enhancer Labrasol^®^ and lipophilic vehicle solubilizer Labrafac^®^ WL 1349. First, Lapatinib (0.35–0.63% *w*/*w*) was dissolved in Labrasol^®^ (10.24–10.51% *w*/*w*) with continuous shaking and heating to a degree below the lapatinib melting point (136–150 °C) to ensure complete dissolution of the drug. Then Labrafac^®^ WL 1349 (4.3% *w*/*w*), Kolliphor^®^ HS15 (7.59% *w*/*w*), Lipoid^®^ S75-3 (0.673% *w*/*w*), NaCl (0.87% *w*/*w*) and water (21.73% *w*/*w*) were added and heated under magnetic stirring up to 85 °C. Three progressive heating and cooling cycles were carried out between 85 °C and 45 °C. At the inversion phase (61–63 °C) of the last cooling cycle, 2 °C deionized water (53.89% *w*/*w*) was added to the mixture. Then the formulation was mixed for extra 5 min under magnetic stirring to form LAPA_LNCs. 

Fluorophore-loaded lipid nanocapsules (DiD_LNCs) were formulated in order to follow the nanocarrier cell uptake. To prepare DiD_LNCs, the DiD dye was dissolved in a ratio of 2% of the core containing both Labrasol^®^ (10.68% *w*/*w*) and Labrafac^®^ WL 1349 (4.35% *w*/*w*) until complete dissolution of the dye. The rest of the protocol remained the same. 

Blank LNCs loaded without any active contents were used as control formulations. For blank LNCs, Labrasol^®^ (10.87% *w*/*w*) and Labrafac^®^ WL 1349 (4.35% *w*/*w*) were directly mixed with the other components, and the heating-cooling cycles were performed as described before. 

#### 3.2.2. Formulation of siRNA (Co-Loaded) Lipid-Based Nanoparticles

A transacylation reaction was carried out between the pegylated hydroxy stearate and the functional amino groups of the chitosan oligosaccharide lactate polymer to obtain LBNP. A method already described by Messaoudi et al. was adapted [33]. Briefly, 20 mL of blank LNCs, LAPA_LNCs, or DiD_LNCs were mixed with 1 mL of NaOH 10 M and three different chitosan concentrations (0.6, 1.2, and 1.8 mg/mL), referred to as chitosan 1×, 2×, and 3×, respectively. The reaction took place at 25 °C in a water bath for 15 min, and afterwards, the reaction was stopped by adding 20 mL of a glycine buffer. Finally, a dialysis-based purification step with membranes having a molecular weight cut-off of 100 KDa was performed for 24 h with Milli-Q water under magnetic stirring. Water was replaced every hour for the first three hours. This allowed transacylated nanoparticles, hereafter called lipid-based nanoparticles (LBNP), to remain inside the dialysis membrane while the free unbound chitosan diffused in the dialysis water. Afterwards, the pH of the final LBNP was adjusted to 1–2 with an HCl solution for a correct ionization of the chitosan before the electrostatic addition of the siRNA. Next, the siRNA solution was added to the LBNP suspension at a volume ratio of 1:3, and the mixture was vortexed for 10 s. The formed siRNA_LBNP, siRNA-LAPA_LBNP, or siRNA-DiD_LBNP were used immediately either for nanocarrier characterization or further cellular experiments. The concentration of the siRNA solution was adapted according to further use.

### 3.3. Physicochemical Characteristics of the Nanocarrier

#### 3.3.1. Particle Size and Zeta Potential

The formulation’s size, polydispersity index, and zeta potential were measured using a Malvern NanoZS instrument (Malvern Instruments, Malvern, UK). The size measurement was performed after diluting the (blank or loaded) LNCs and LBNP suspension by a factor of 10 and 60, respectively, in Milli-Q water at 25 °C. All formulations had comparable conductivity values for zeta potentials. All measurements were done in triplicate.

#### 3.3.2. Encapsulation Efficiency 

The amount of LAPA encapsulated in LNCs was determined immediately after LAPA_LNCs formulation, after LAPA_LBNP purification, and after pH adjustment to pH 1–2 to ensure the different formulation steps did not affect LAPA_LBNP integrity. Each batch was filtered using a polyether sulfone^®^ 0.2 μm filter (Clearline, D. Dutscher, Brumath, France) to remove free LAPA from LNCs/LBNB suspension. Three samples of each batch (filtrated and non-filtrated) were prepared by dissolving the LAPA-loaded LNCs/LBNP (125 μL) with an equivalent volume of water and THF (1 mL). Afterwards, a UV-visible spectrophotometer measured the solution’s absorbance between 250 and 400 nm (Genesys 10S, Thermo Scientific, France). Quantification was achieved by comparing LAPA absorbance at 335 nm to a calibration curve made with blank nanocarriers and LAPA/THF/water mixture. Drug loading (DL = amount of LAPA per weight of LNCs suspension; mg/g) and encapsulation efficiency (EE) (%) were calculated using the following formulas. The concentration of LAPA (Mw = 581.1 g/mol) in LNCs suspension was also expressed in molarity (the density of the LNCs suspension is considered the same as the density of water).
DL (mg/g) = Weight of LAPA in LNCs suspension (mg)/weight of LNCs (g)
EE (%) = _Encapsulated_ Lapatinib/_Total_ Lapatinib × 100

#### 3.3.3. Agarose Gel Electrophoresis

Agarose gel electrophoresis assay was performed to check the complexation of siRNA into the nanocarrier. Samples were prepared in order to have a final siRNA concentration of 1.2 µM per well. To control the integrity of the formulations and release siRNA, samples were used in the presence and absence of heparin (final concentration of 3 mg/mL) (Sigma-Aldrich Chemie GmbH, Steinheim, Germany). A loading buffer (2X RNA loading dye, Life Technologies, Paisley, United Kingdom) was added to the samples before loading them into the wells. An agarose gel (1% *m*/*v*) was prepared by dissolving Agarose (Low-EEO/Multi-Purpose, Acros Organics BV, Geel, Belgium) in Tris-acetate-EDTA (TAE) solution 1X (Acros Organics BV, Geel, Belgium) containing 0.01% (*v*/*v*) ethidium bromide (EtBr) to visualize free siRNA. After deposition of the samples on the agarose gel, the electrophoresis migration was conducted in TAE 1X buffer for 15 min at 150 V. The gels were visualized by UV-imaging using the EvolutionCapt software on a Fusion-Solo. 65.WL imager (Vilbert Lourmat, Marne-la-Vallée, France).

#### 3.3.4. Storage Stability of the Nanoparticles

First, blank LBNPs with three different chitosan concentrations (1×, 2×, and 3×, respectively) were stored at a low temperature (4 °C) and protected from light. On days 0, 7, 14, 21, and 28, the particle size, PDI, and zeta potential were measured after the siRNA addition. Then, a siRNA complexation assay was performed using gel agarose electrophoresis. These experiments were carried out to investigate the stability of blank LBNPs and their ability to conserve the physicochemical characteristics and siRNA loading over time (more information about the comparison between the three formulations is mentioned in the Appendix A.

Optimized LBNP with 3× chitosan concentration (1.8 mg/mL) underwent storage stability investigation utilizing siRNA_LBNP and free siRNA-chitosan prepared by a physical mixing with water. In this case, the siRNA was added on day 0 and tracked over time. All the formulations were kept as mentioned above for up to 28 days at 4 °C and characterized in the same way on days: 0, 7, 14, and 28.

### 3.4. Nanocarrier Cellular Evaluation

#### 3.4.1. Cell Line and Culture

SK-BR-3 human breast carcinoma cell lines with HER2 overexpression were purchased from Cell Lines Service (CLS Eppelheim, Germany). SK-BR-3 cells were grown at 37 °C, and 5% CO_2_ in Dulbecco’s Modified Eagle Medium (DMEM) supplemented with 10% fetal bovine serum (FBS, Gibco^®^) and 1% Penicillin–Streptomycin solution (10,000 U/mL, Gibco^®^).

#### 3.4.2. Confocal Spectral Imaging (CSI)

For confocal spectral imaging, ATTO488 siRNA-DiD_LBNP distribution was analyzed on cell-adherent coverslips. Cover glasses treated with poly-D-lysine were placed in 24-well plates. They were seeded with 3 × 10^4^ SK-BR-3 cells and placed for 48 h in a culture medium. The cells were then incubated with ATTO488 labeled siRNA-DiD_LBNP (150 nM final siRNA concentration) in OptiMEM for 4 h and washed three times with PBS. The cover glasses were placed between a microscope slide and a coverslip to be scanned for CSI using a LabRAM laser scanning confocal microspectrometer (Horiba SA, Villeneuve d’Ascq, France) equipped with a 300‘/mm diffraction grating and a CCD detector air-cooled by Peltier effect. The DiD fluorescence was excited using a 633 nm line of a built-in He-Ne laser, and the fluorescence of ATTO488 was excited with a 491 nm laser under a 50X long focal microscope objective. The laser light power at the sample was approximately 0.1 mW, and the acquisition time was 50 ms per spectrum. For the analysis of adherent cells, an optical section (x-y plane) situated at half-thickness of the cell was scanned with a step of 0.8 μm that provided maps containing typically 2500 spectra. Both acquisition and treatment of multispectral maps were performed with LabSpec software version 5.

#### 3.4.3. In Vitro Cytotoxicity

Cell viability and proliferation were studied using a luminescent test based on the quantification of ATP using the CellTiter-Glo cell proliferation assay (Promega, Madison, WI, USA). Briefly, 6000 cells of SK-BR-3 were incubated in 100 μL of the medium in 96 well plates for 24 h and then treated with concentrations ranging from 0.01 nM to 100 µM of tested compounds. An H_2_O_2_ solution at 20 mM was used as a positive control, and the culture medium alone was tested as a negative control. LAPA ditosylate salt, LBNP loaded LAPA with either Ctrl. siRNA or anti-Survivin siRNA and blank LBNP were tested on cells. Cells were incubated with 100 μL of each solution at 37 °C with 5% CO_2_ for 4 days. Cell viability was then determined using the Cell Titer-Glo reagent (Promega, Madison, WI, USA). Briefly, 100 µL of the medium was removed, and 100 μL of Cell Titer-Glo reagent was added to each well. The plates were shaken for 2 min and then incubated at room temperature for 10 min. The luminescence values were measured with an acquisition at 0.5 s, using an absorbance microplate reader (Bio-Tek^®^ instruments, Inc., Winooski, VT, USA). When a dose-dependent activity was observed, 50% inhibitory concentration (IC_50_) was calculated using Graphpad PRISM 7 software (*n* = 4 in quadruplicate).

### 3.5. Statistical Analysis

For physicochemical characterization, all the formulations were repeated at least three times. All data are presented as mean ± standard deviation. IC_50_ and *p* values were calculated using GraphPad PRISM 7 software.

## 4. Conclusions

In sum, the current study investigated the potential of the co-delivery of LAPA and anti-Survivin siRNA in modified lipid nanocapsules to provide a synergistic therapeutic effect on HER2 overexpressed SK-BR-3 cell line. The siSurvivin-LAPA_LBNP exhibited suitable physicochemical properties as a parenteral delivery system. The combination effectively inhibits cell proliferation and induces cell apoptosis and marked inhibition of Survivin protein expression. The favorable anti-cancer effect was attributed to the synergistic effect of LAPA efficacy and apoptotic induction maintained via Survivin protein knockdown in vitro. Although this synergy was not as high as predicted, our findings support the growing evidence that siRNA treatment combined with anti-cancer drugs represents a new modality in treating one of the aggressive types of breast cancer (HER2+). In the future, this nanocarrier could be further modified and tested using an in vivo model with different protein targets involved in regulating genes responsible for MDR, apoptosis, and many cancer cell survival pathways.

## Figures and Tables

**Figure 1 pharmaceuticals-15-01452-f001:**
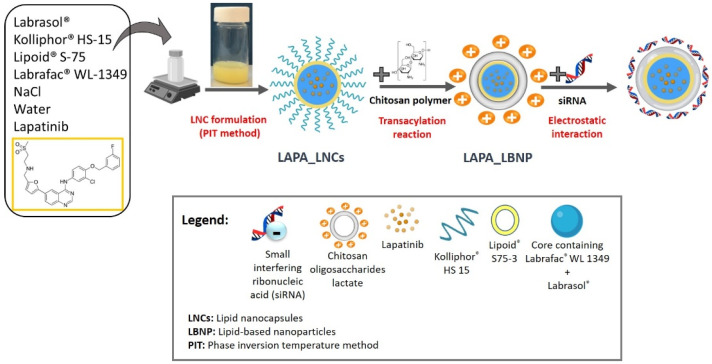
Schematic illustration of the siRNA-LAPA_LBNP formulation process.

**Figure 2 pharmaceuticals-15-01452-f002:**
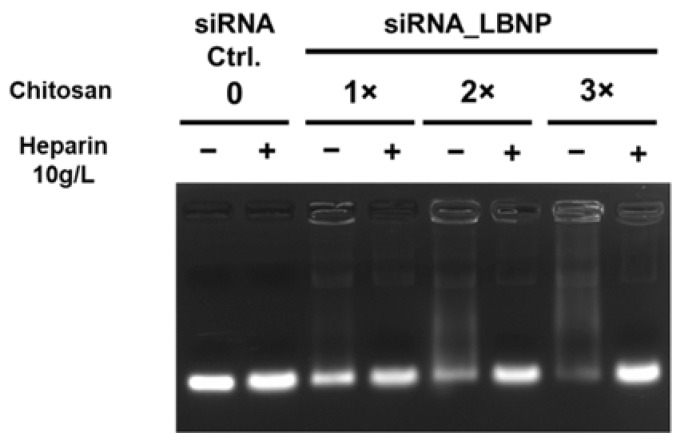
Gel retardation assay image demonstrating the siRNA protection in the modified lipid-based nanoparticles with different chitosan concentrations after formulation (day 0). The initial theoretical chitosan concentration of 0.6 mg/mL is 1×, while the double and triple chitosan concentrations are 2× and 3×, respectively. siRNA formulated in LBNP in the presence (+) or absence (−) of heparin was compared to naked siRNA. Lanes without heparin show free siRNA amount, and lanes with heparin show total siRNA amount in the sample.

**Figure 3 pharmaceuticals-15-01452-f003:**
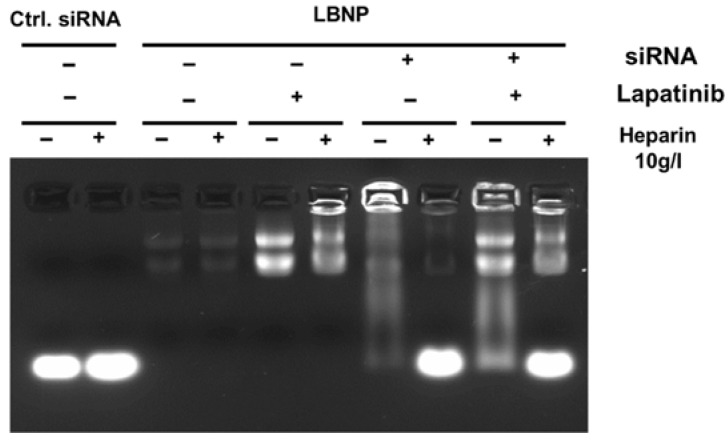
Gel retardation assay image demonstrates siRNA protection in the modified lipid-based nanoparticles. Control siRNA formulated in blank or Lapatinib loaded LBNP in the presence (+) or absence (−) of heparin were compared to naked siRNA. Lanes without heparin show free siRNA amount, and lanes with heparin show total siRNA amount in the sample.

**Figure 4 pharmaceuticals-15-01452-f004:**
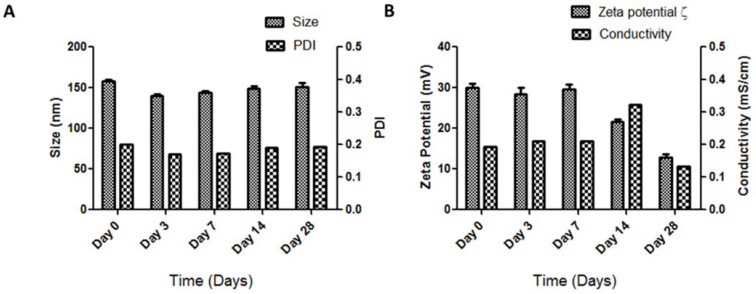
The graphs represent the stability study based on physicochemical parameters of the modified lipid-based nanoparticles with siRNA added before each measurement on days: 0, 3, 7, 14, and 28, indicated by (**A**) Size and polydispersity index (PDI) and (**B**) Zeta potential and conductivity values. The theoretical chitosan concentration was (1.8 mg/mL) which was the optimized chitosan concentration used for all experiments. The blank LBNP storage temperature was 4 °C.

**Figure 5 pharmaceuticals-15-01452-f005:**
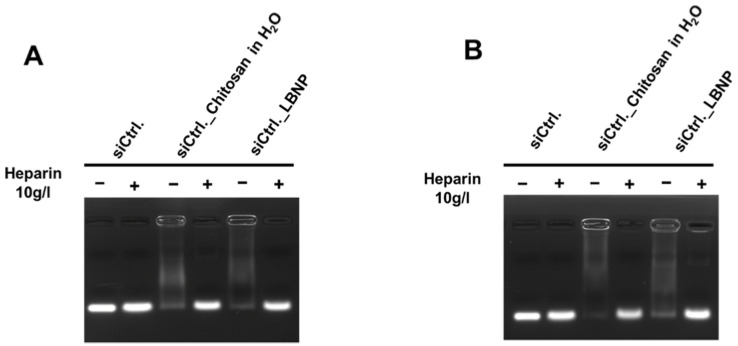
Gel retardation assay images demonstrate siRNA protection of siCtrl._Chitosan in H_2_O (pH < 5) and siCtrl._loaded LBNP on day 0 (**A**) and day 28 (**B**). The theoretical chitosan concentration was 1.8 mg/mL (optimized concentration). siRNA formulated in each condition in the presence (+) or absence (−) of heparin was compared to naked siRNA. Lanes without heparin show free siRNA amount, and lanes with heparin show total siRNA amount in the sample. The ctrl. siRNA was added once on day 0.

**Figure 6 pharmaceuticals-15-01452-f006:**
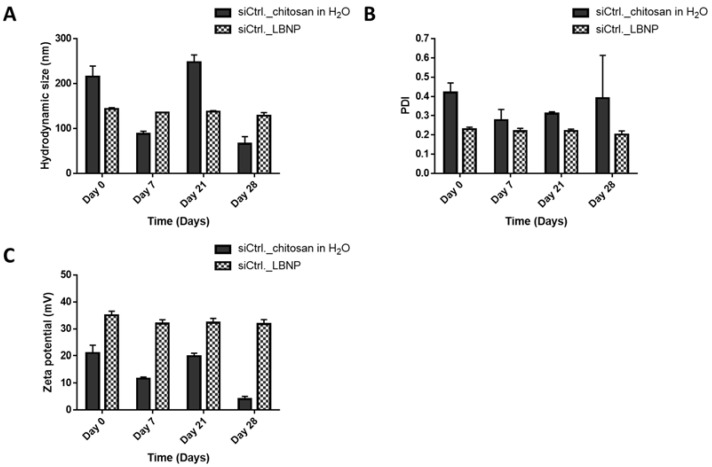
Stability study graphs of size (**A**), polydispersity index (PDI) (**B**), and zeta potential (**C**) variation over the time of siCtrl._Chitosan in H_2_O (Polyplex) (pH < 5) and siCtrl._loaded LBNP. The formulation’s storage temperature was 4 °C. The ctrl. siRNA was added once on day 0.

**Figure 7 pharmaceuticals-15-01452-f007:**
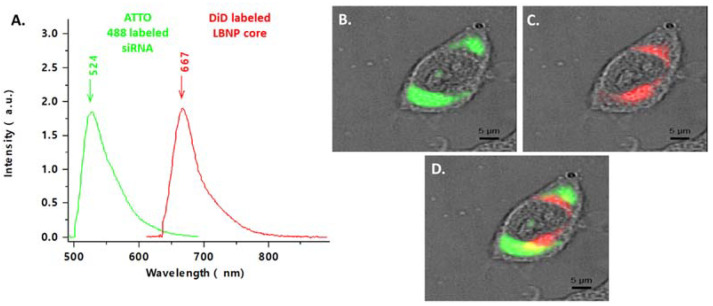
Uptake of fluorescent LBNP and fluorescent siRNA in SK-BR-3 cells after 4 h incubation. Cells were analyzed by confocal spectral imaging, DiD labeled LBNP are represented in red, and ATTO-488 labeled siRNA loaded LBNP in green. Representative spectra of both fluorochromes (**A**), representative images of each fluorochrome alone (**B**,**C**), and merged (**D**).

**Figure 8 pharmaceuticals-15-01452-f008:**
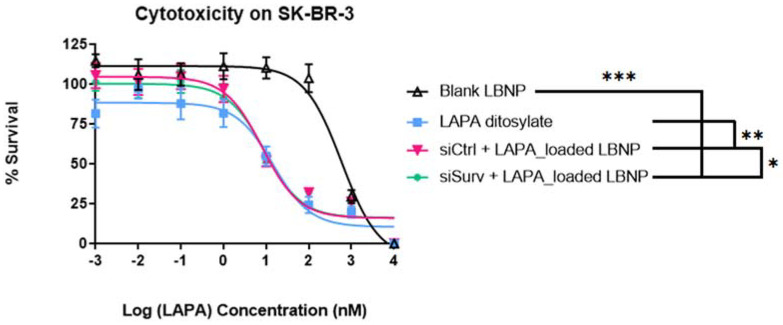
Dose-response curves of blank, siCtrl.-LAPA and siSurvivin-LAPA-loaded LBNP formulations; and Lapatinib ditosylate as a free drug. The half-maximal inhibitory concentration (IC_50_) values were calculated from cell survival curves in SK-BR-3 cells after 96 h incubation. * *p* value < 0.05 (significant), ** *p* value < 0.01 (highly significant), *** *p* value <0.001 (extremely significant).

**Table 1 pharmaceuticals-15-01452-t001:** Mean particle size, polydispersity index, and zeta potential of the LNCs, LAPA_LNCs, LAPA_LBNP, and siRNA-LAPA_LBNP.

Formulation	D_H_ (nm)	PDI	Zeta Potential (mV)	EE (%)
LNCs	86.9 ± 12.9	0.116 ± 0.02	−4.15 ± 4.35	-
LAPA_LBNP	126.9 ± 20.60	0.14 ± 0.08	+28.42 ± 6.69	94.51 ± 6.63
siRNA-LAPA_LBNP	123.9 ± 17.10	0.09 ± 0.05	+20.84 ± 8.67	

D_H_: Hydrodynamic diameter, EE: Encapsulation efficiency, PDI: Polydispersity index, LAPA: Lapatinib, LBNP: Lipid-based nanoparticles, LNCs: lipid nanocapsules, mV: millivolt.

**Table 2 pharmaceuticals-15-01452-t002:** The IC_50_ values of different formulations of LBNP and free LAPA.

Tested Formulations	IC_50_ (nM)
LBNP	6481 ± 1486
LAPA_ditosylate	159.0 ± 12.4
siCtrl.-LAPA_LBNP	99.7 ± 12.8
siSurv.-LAPA_LBNP	76.8 ± 12.3

LBNP: Lipid-based nanoparticles, LAPA: Lapatinib, siCtrl.: Control siRNA, siSurv.: anti-Survivin siRNA, IC_50_: Half-maximal inhibitory concentration.

**Table 3 pharmaceuticals-15-01452-t003:** Actual *p* values of control and tested formulations calculated based on IC_50_ variations obtained from the cytotoxicity assay.

Formulations		Statistical Significance
Control	Evaluated/Compared	*p*-Value
LBNP	siCtrl.-LAPA_LBNP	0.0001 ***
	siSurv.-LAPA_LBNP	
	LAPA Ditosylate	
siCtrl.-LAPA_LBNP	LAPA Ditosylate	0.0006 ***
siCtrl.-LAPA_LBNP	siSurv.-LAPA_LBNP	0.0418 *

LBNP: Lipid-based nanoparticles, LAPA: Lapatinib, siCtrl.: Control siRNA, siSurv.: anti-Survivin siRNA. * *p* value < 0.05 (significant), *** *p* value <0.001 (extremely significant).

## Data Availability

Data is contained within the article and Appendix A.

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
