# Peer review of "Formulation of Lipid-Based Nanoparticles for Simultaneous Delivery of Lapatinib and Anti-Survivin siRNA for HER2+ Breast Cancer Treatment"

_pharmaceuticals, 2022, doi:10.3390/ph15121452_

Round 1

Reviewer 1 Report

The work was well done, and it was reported clearly. The experiments were well designed and described. The methods were appropriate, and the presented data supported the conclusions.

I have a few comments:

Line 41 to 51 need more citation

Sometimes word Figure was written in italic. Also, in the supplementary file, please be consistent with writing the figure number and the font size(for example, Fig.S1, Fig. 2S).

Page 4 – Line 145, (Fig. S1,4) There is no Fig S4

Figure 2 and Fig.S2 look similar!!!, No data shows significant siRNA release after 7 days.

Why did the Zeta potential of siCtrl.-DiD_LBNP in Table S1 become negative?

Please enhance the resolution of Figure 7.

Page 11- line 402, citation style did not match the journal style.

Author Response

EAV

Reviewer 2 Report

Although the authors said that they would present the research results for the treatment of Her2+ breast cancer tumors through the preparation of nanoparticles containing lapatinib and siRNA. This manuscript presented only cytotoxicity results in SK-BR-3 cells. The results presented by the authors are mostly raw results of the process of manufacturing nanoparticles, and there is no scientific novelty or inventive step. Therefore, I think that this manuscript cannot be published in pharmaceuticals.

1.     In section 2.1., please add a description of the composition and preparation method of LAPA-LBNP. Please clearly explain the contents of Figure 1 in the main text.

2.     The authors explain "The final formulations of LNCs and LAPA_LNCs were ~ because of the Tyndall effect". Please provide clear results.

3.     In Section 2.2., Please check the Figure label “Fig. S1, 4).

4.     The image in Figure S2 well blurs proper judgment. It should be well established. Additionally Figure 2S should be modified to S2.

Reviewer 3 Report

Proposed manuscript is good piece of research work, following are some minor revisions should be taken in consideration

1. Introduction can be improved by focusing on cancer targeting. Also, the techniques other than lipid-based technologies can be the part of introduction for example Cyclodextrin NPs. For that refer and cite following article

       https://doi.org/10.1016/B978-0-12-822351-2.00014-0

2. Check the grammar and sentence errors and rectify

3. Improve the resolution of all the figures

Reviewer 4 Report

The authors described the co-delivery of lapatinib/siRNA(surviving) for HER2+ breast cancer treatment using lipid nanocapsules (LNCs), and they mainly conducted physicochemical characterization of the LNCs. However, the concept of co-delivery of anticancer drugs and siRNA (or antisense oligo) does not have impacts. The obtained in vitro results are apparently no different though they showed statistical differences. 

Round 2

Reviewer 2 Report

The manuscript has been well revised and can be published without further revisions.

Reviewer 4 Report

I could not find any scientific significance in this revised paper.

Most of this paper has focused on the characterization and stability of their nanoparticle, which is not new. Besides, the in vitro outcome, IC50 difference between siSurv and siCtrl, is little (76.8±12.3 and 99.7±12.8) even if they have statistical differences. There are a lot of effective siRNA/anticancer drug co-delivery systems. Compared to them, this system is obviously less effective.

Figure 7 showed a single cell, and there is no quantitative data for cellular uptakes, such as FCM. The several cells in a single image and quantitative data must be required.

The authors used only one cell line (SK-BR-3) overexpressing the HER2 gene. If the authors want to claim the delivery to HER2+ breast cancer treatment in the title, they should use HER2– cell line as a negative control.

In addition to Western blot, downregulation of the target surviving mRNA by RNAi must be needed, if the authors clearly show siRNA delivery.

Line 173: The authors mentioned that “the surface charge of blank LNCs bears a neutral or moderate negative surface charge due to the presence of the PEG shell”. However, PEG itself bears no negative charges. Thus, they need to alter and/or add descriptions.